# Serial founder effects slow range expansion in an invasive social insect

Thomas Hagan [1] ✉, Guiling Ding [1,2], Gabriele Buchmann [1], Benjamin P. Oldroyd[1] & Rosalyn Gloag [1] ✉

Invasive populations often experience founder effects: a loss of genetic diversity relative to the source population, due to a small number of founders. Even where these founder effects do not impact colonization success, theory predicts they might affect the rate at which invasive populations expand. This is because secondary founder effects are generated at advancing population edges, further reducing local genetic diversity and elevating genetic load. We show that in an expanding invasive population of the Asian honey bee (*Apis cerana*), genetic diversity is indeed lowest at range edges, including at the *complementary sex determiner*, *csd*, a locus that is homozygous-lethal. Consistent with lower local *csd* diversity, range edge colonies had lower brood viability than colonies in the range centre. Further, simulations of a newly-founded and expanding honey bee population corroborate the spatial patterns in mean colony fitness observed in our empirical data and show that such genetic load at range edges will slow the rate of population expansion.

Invasive species are a major threat to natural and agricultural ecosystems around the globe where they act as competitors, predators, parasites or pathogen vectors for resident species[1]. Given this ecological and economic impact, a key goal is to understand the eco-evolutionary processes that affect the odds of invaders establishing and their population dynamics once established. One such major process is the founder effect, which describes the reduction in genetic diversity and change in allele frequencies in an invasive population, relative to the parent population, due to a small number of founding individuals[2]. A small founding population may not reflect the allelic composition of its source population because founding individuals fail to transfer all source alleles to the new population and/or because founder allele frequencies differ strongly from those of the source population[3]. These effects are then exacerbated by genetic drift in the generations immediately following the founding event[4]. The potential implications of founder effects for newly invasive populations have long been recognized, and include inbreeding depression, reduced adaptive capacity and/or high extinction risk[5–8]. Less well recognized, however, is the possibility that invasive populations may also experience secondary, localized founder effects at

range edges following the initial post-colonizing founder effect. These occur because limited numbers of individuals disperse outwards and mate, repeatedly founding "new" edge populations[9,10]. Such secondary founder effects could also be important for population dynamics as they exacerbate the effects of already lowered genetic diversity in a population, increasing homozygosity and reducing fitness at range edges[10,11], which in turn curbs the rate of range expansion[12].

While secondary founder effects at range edges are predictable via a verbal model as outlined above, their demonstration in practice is challenging[13,14]. This is because such demonstrations are contingent on our ability to identify fitness-relevant genotypes and map their spatial variation at population scales. Invasive hymenopterans in the clade Aculeata (the ants, bees, and stinging wasps), however, provide an opportunity to do so due to their system of sex determination. Hymenopterans are haplodiploid: males are haploid, and females are diploid. Yet in many species, sex is ultimately determined by zygosity at one or more "sex loci" in a system known as Complementary Sex Determination[15,16]. All haploid individuals develop as normal males and diploid individuals develop as females if they are heterozygous at the

[1]Behaviour, Ecology and Evolution Lab, School of Life and Environmental Sciences, University of Sydney, Sydney, NSW 2006, Australia. [2]Key Laboratory of Pollinating Insect Biology of the Ministry of Agriculture and Rural Affairs, Institute of Apicultural Research, Chinese Academy of Agricultural Sciences, Beijing 100093, China. ✉e-mail: thmh6@umsl.edu; ros.gloag@sydney.edu.au

sex locus (or at least one sex locus if there are multiple). Diploid individuals that are homozygous at the sex locus (or loci) instead develop as "diploid males" that are infertile, inviable or cannibalised as larvae[15]. For social species, diploid males may be additionally costly as they take the place of female workers needed for colony function[17,18]. In large established populations, strong heterozygote advantage (balancing selection) at sex loci maintains a high number of distinct alleles, keeping heterozygosity high and the frequency of diploid males low[19]. However, in invasive populations that have suffered population bottlenecks, loss of allelic diversity results in higher incidences of diploid males relative to native-range populations[20–22]. This is because when there are fewer distinct sex alleles in a population, it is more likely that mates share the same sex-determining allele and thus that their offspring are homozygous (i.e., diploid males). At fitness-critical sex loci, therefore, we predict that secondary, serial founder effects at range edges will impact the rate of expansion of invasive hymenopteran populations.

Here we show how genetic diversity and brood viability vary with distance from population range edges in an expanding invasive honey bee population. The Asian honey bee (*Apis cerana*) is native to Asia but became established in northern Australia following the accidental introduction of just one, or very few, colonies around 2007[23,24]. In a previous study[25] we assessed changes in sex allele number and frequency in this population during the first eight years post-colonization and showed that the initial founding bottleneck reduced allele diversity at the honey bee's single-sex locus (named the *complementary sex determiner*, *csd*[26]) by at least two-thirds (7 *csd* alleles vs. >20 in native-range local populations[25]). As expected, the population retained greater diversity at *csd* than neutral loci over time, because strong frequency-dependent (balancing) selection limited the loss of rare

alleles due to genetic drift. Selection also drove *csd* allele frequencies towards equal frequency over time, reducing the high incidence of diploid male production caused by allele frequency skew after the founder event. Australia's *A. cerana* population, however, continues to expand its range, providing an opportunity to look for evidence of serial founder effects at range edges. We show that the same high allelic skew at *csd*, and corresponding high genetic load of the initial founder event, now plays out on repeat at range edges as the population expands. We then complement this empirical data with simulations that show how genetic load at the sex locus reduces the population's rate of spread, relative to a population where no such genetic load occurs.

## Results
### Genetic diversity at range edges
We first assessed how population genetic diversity at both the sex locus (*csd*) and neutral loci varied with distance from the range centre in Australia's invasive *A. cerana*, using drones sampled at drone congregation areas (DCAs; Fig. 1). DCAs provide an efficient method of sampling the genetic diversity of a local honey bee population as they attract males from all colonies within an approximately 3.75 km radius[27]. Moreover, each aggregation of males represents the gene pool available to queens that visit that aggregation to mate. We found significant spatial heterogeneity across the population in *csd* frequencies, with the Northern and Southern Range Edge regions having significantly reduced haplotype diversity (*H*) relative to the overall population ($p_{Northern\ Edge} < 0.01$, $p_{Southern\ Edge} < 0.001$; Bootstrapped Monte Carlo Simulations, see "Methods" section and Fig. 1B; $N_{TOTAL} = 4639$). This reduction in *csd* haplotype diversity (the result of a shift in allele frequencies relative to the central region), was modest

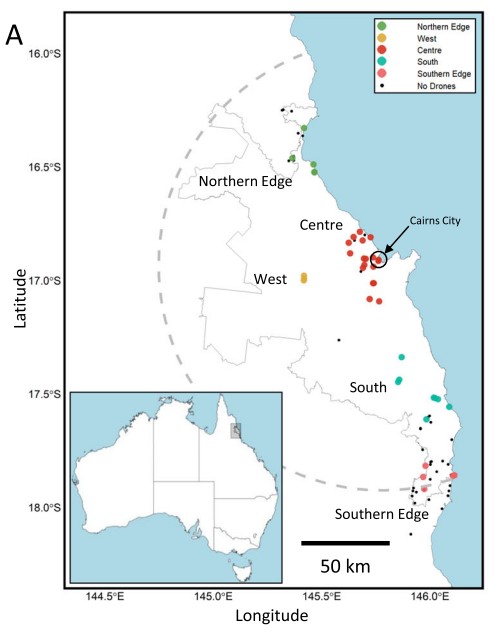
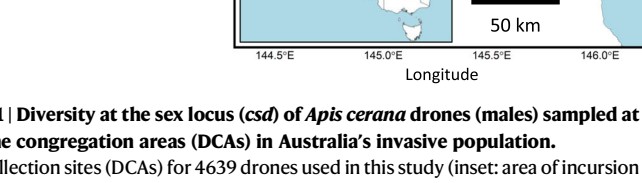
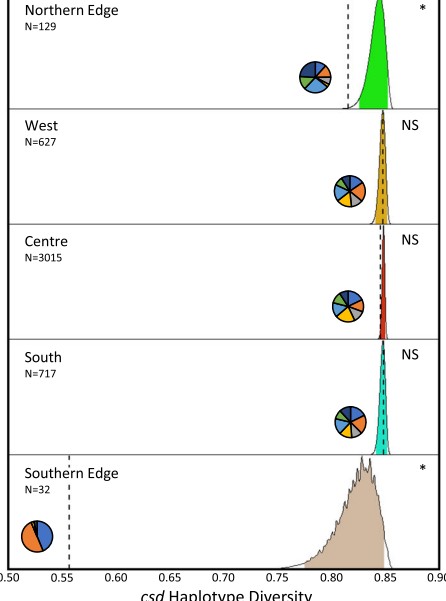

**Fig. 1 | Diversity at the sex locus (*csd*) of *Apis cerana* drones (males) sampled at drone congregation areas (DCAs) in Australia's invasive population.**
**A** Collection sites (DCAs) for 4639 drones used in this study (inset: area of incursion in North-East Australia). Sites are coloured by region within this continuous population (Northern = Green, West = Orange, Centre = Red, South = Cyan and Southern Edge = Pink). Points in black denote DCAs where we failed to collect drones; these empty DCAs were more common at range edges than the range centre, consistent with lower population density at range edges. The predicted range edge is shown with a grey dashed line, and the known infested range of *A. cerana* based on collections by Queensland Biosecurity is shown with a solid grey

line (year = 2019). **B** The results of Monte Carlo simulations of sex locus (*csd*) allele diversity in each region, with sample sizes per region (N) and haplotype diversity being denoted on the x-axis. Dashed lines represent the observed haplotype diversity, and pie charts represent the frequency per region of the seven *csd* alleles present in this population. Distributions show the bootstrapped range of haplotype diversities for the region when sampled from the allele frequency of the entire population, with shaded regions representing the most common 95% of haplotype diversities. Where the dashed lines fall outside of this shaded region (northern and southern edges), the *csd* haplotype diversity of the region is significantly different from that in the overall population.

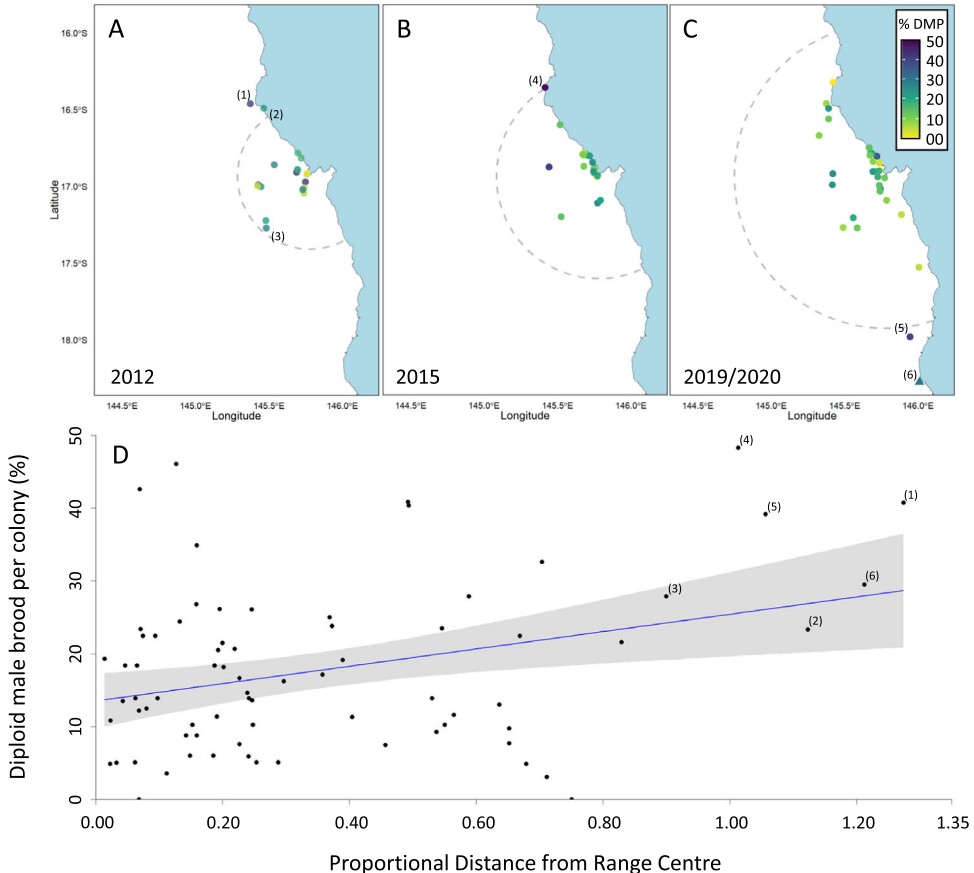

**Fig. 2 | The relationship between diploid brood viability and distance from the range centre. A–C** Locations of colonies collected in an expanding invasive population of *Apis cerana* in 2012, 2015 and 2019/2020 colour-coded by the relative proportion of diploid brood (DMP) per colony that are inviable 'diploid males' (0–50%; yellow-blue). The estimated range edge of the population per year (2012, 2015 and 2020) is shown by the dotted line. In total, six colonies (1–6) were sampled at range edges: three at the northern edge, two at the southern edge and one at the western edge (note (6) was collected in July 2022). **D** The proportion of diploid male brood produced per colony and the distance the colony was found from the centre of the population (*n* = 74 colonies). Distance is represented as a proportion of the maximum distance between the predicted range edge and the point of incursion (range centre) at the time of colony collection. The grey regions surrounding the regression line indicates the 95% CI of regression. Range edge colonies 1–6 are indicated.

at the Northern Edge (*H* = 0.82 vs 0.85 in the range centre) but severe at the Southern Edge (*H* = 0.56).

A similar pattern was evident at neutral loci based on the nucleotide diversity (π) at 2829 genome-wide SNPs (*N* = 63 drones from DCAs in all subregions except West; 15–16 per region), with nucleotide diversity highest in the Centre (π = 2.22 × 10⁻³) and South (π = 2.23 × 10⁻³) regions, followed by the Northern Edge (π = 2.11 × 10⁻³) and then the Southern Edge (π = 2.00 × 10⁻³); Supplementary Table 1. Moreover, this SNP dataset indicated genetic differentiation between regions consistent with advancing edge founder effects. Pairwise $F_{st}$ values increased as the distance between regions increased (Centre vs South: $F_{st}$ = 0.049, Centre vs Northern Edge: $F_{st}$ = 0.065, Centre vs Southern Edge: $F_{st}$ = 0.095, Northern Edge vs Southern Edge: $F_{st}$ = 0.112; Supplementary Table 2) and a principal component analysis (PCA) showed some Southern Edge and Northern Edge individuals falling outside of the main population cluster along Principal Components 1 and 2 respectively (these PCs cumulatively explained 10.9% of the variation in the population; Supplementary Fig. 1A). Haplotype diversity at two of eight microsatellite (single sequence repeat, SSR) loci was also reduced at the Southern Edge relative to the Centre (Ac3: $H_{Southern\ Edge}$ = 0.06, $H_{Centre}$ = 0.4; Ac27: $H_{Southern\ Edge}$ = 0.06, $H_{Centre}$ = 0.25; 4643 drones; Supplementary Table 3) though no such reduction was evident from microsatellite markers at the Northern Edge. The low overall polymorphism at microsatellite loci across the population meant that they had limited power to detect within-population spatial differentiation (2–4 alleles per locus; average polymorphic information content = 0.31; pairwise $F_{st}$ values based on SSRs <0.07 in all cases; Supplementary Tables 3–4; Supplementary Fig. S1B; $N_{TOTAL}$ = 4639 drones).

### Brood viability (diploid male production) at range edges

The proportion of eggs in a colony that are inviable diploid males can be determined from the *csd* genotypes of the colony's workers (see "Methods" section and Supplementary Fig. 2). Based on the theory of serial founder events, and on our population genetic diversity data, we predicted that the proportion of diploid male brood per colony would be elevated towards range edges relative to the central population. To test this prediction, we calculated the proportion of diploid male brood in colonies sampled throughout the invasive range in the years 2012, 2015 and 2020–2022 (Fig. 2A–C; *N* = 74 colonies). Consistent with the results from spatial variation in sex allele frequencies observed in the drones, we found a significant positive relationship between the distance a colony was found from the range centre, *x*, and the proportion of diploid male brood it was producing, *y*, (*y* = 0.12× + 0.14; $R^2$ = 0.09, $F_{1,72}$ = 7.415, *p* < 0.01, linear regression; Fig. 2D). Thus while colonies with high diploid male production still occurred in the population centre, on average, those colonies closer to the range edges had higher proportions of inviable diploid male brood

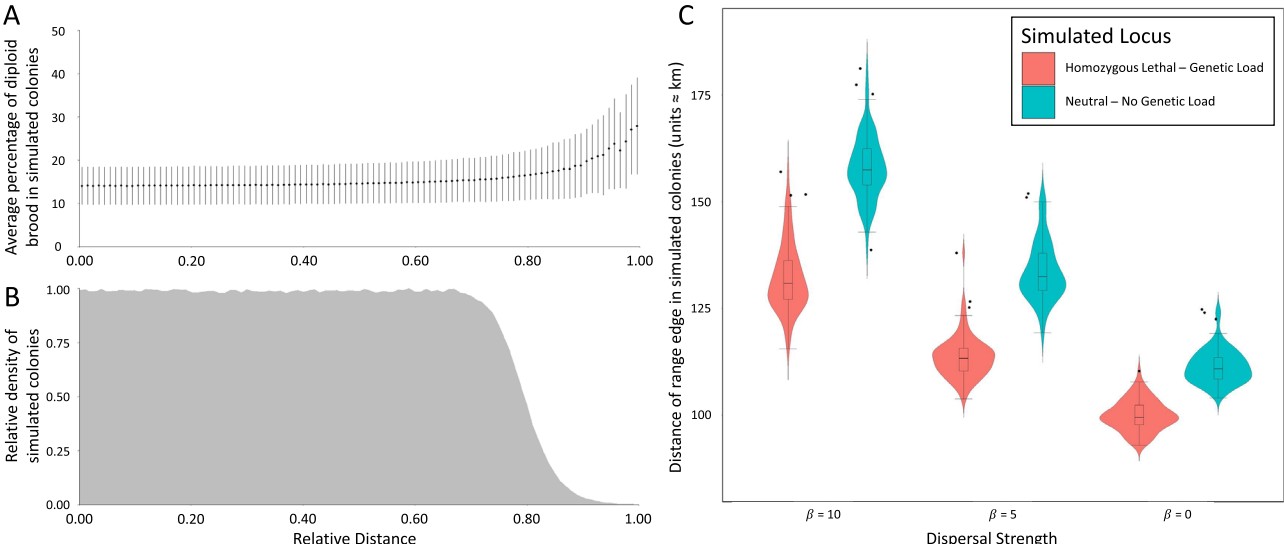

**Fig. 3 | The results of agent-based simulations of an invading and dispersing honey bee population. A** The relationship between the average proportion of diploid male brood of simulated colonies and the distance these simulated colonies are found from the centre of the population (Sigmoid fitness function, $K = 100$, $S_A = 3$ and $\beta = 5$, $n = 96$). Here each point represents the average fitness of colonies binned at that percentile of distance for each simulation, and error bars represent standard deviation in fitness. **B** The relationship between colony density and the distance colonies are found from the population centre in our simulations (Sigmoid fitness function, $K = 100$, $S_A = 3$ and $\beta = 5$, $n = 96$), showing that high diploid male production occurs as colony density decreases at range edges. Distance in both **A** and **B** is represented as a proportion of the maximum range edge the population

reached during the simulation. **C** The effect of genetic load at *csd* on maximum range expansion in simulated populations. Simulations modelling a homozygous-lethal sex locus (ie, *csd*) and a neutral locus are coloured red and blue respectively. Boxplots here show interquartile ranges (IQR; outliers fall outside the whiskers defined by the range: mean ± 2.5*IQR), and violin plots show distributions of range edge distances ($n = 96$ for each group). For maximum dispersal strength ($\beta = 10$), the difference between means is 25.8 units (95% confidence interval is [23.6, 28.0]). For medium dispersal strength ($\beta = 5$), the difference between means is 20.1 units (95% confidence interval is [18.4, 21.8]). For lowest dispersal strength ($\beta = 0$), the difference between means is 11.4 units (95% confidence interval is [10.3, 12.5]).

(Fig. 2D). Indeed, all six colonies sampled at or very near the most extreme range edges (northern edge: $N = 3$, southern edge: $N = 2$, western edge: $N = 1$) had diploid male production in the range 20–50% (i.e. at least one in every five diploid embryos were inviable, and as much as one in every two; mean proportion of diploid male brood for edge and non-edge colonies = 35% and 16% respectively). Queens at range edges were thus more likely to mate with males that shared their sex alleles, presumably because they were frequently mating with brothers or other relatives.

### Simulations of range edge founder effects and their impact on expansion rate

To understand how serial founder effects at *csd* are impacting the rate of spread of Australia's invasive *A. cerana*, we generated agent-based simulations of an invading and dispersing honey bee population. This model used a single, continuous spatial dimension where agents reproduced and dispersed in discrete generations. We considered scenarios in which new colonies had some tendency to disperse towards regions of lower density and scenarios in which there was no such tendency (defined by values of a parameter $\beta > 0$ and $\beta = 0$ respectively). As our simulated populations grew and spread, *csd* allele frequencies in the centre of populations became more equal, as expected under frequency-dependent selection. However, skew in allele frequencies persisted at the periphery of the ever-moving range edge. Thus, we found that colonies at the range edge of our simulated populations produced fewer viable brood on average than colonies in the range centre (Fig. 3), consistent with our empirical data. This general pattern held irrespective of whether we assumed a linear or sigmoid relationship between fitness and diploid male brood incidence, and across a range of parameter values for reproductive rate, maximum population density and dispersal strength (Supplementary Methods 1.1). The magnitude of difference in brood viability between edge and centre colonies was higher (and closer to that observed in

empirical data; Fig. 2D) if we assumed new colonies were somewhat more likely to move into unoccupied habitat (away from the range centre) than occupied habitat (towards the range centre), as would occur if new habitat presented less competition for nesting cavities ($\beta > 0$). However, brood viability was always lowest at range edges even if we assumed no such directional preference ($\beta = 0$; Supplementary Methods 2.1).

We then repeated our simulations but with the proportion of diploid male brood per colony artificially set to zero, regardless of the queen and her mates' alleles. That is, we simulated a "null model" population in which the simulated locus is fitness neutral and there is no fitness cost of homozygosity. We then compared the rate of spread for populations under this fitness-neutral condition to that in our previous simulations, by comparing the distances the furthest colonies reached over the life of the simulations (20 generations). We found that populations with no genetic load at the simulated locus reached greater distances on average (that is, expanded range more quickly) than those in which the locus was homozygous-lethal (Fig. 3C). This was true even when founder effects at range edges were rather weak ($\beta = 0$, Difference in means = 11.4 units; $\beta = 5$, Difference in means = 20.1; $\beta = 10$, Difference in means = 25.8; Fig. 3C). Thus, we conclude that genetic load at the sex locus experienced at advancing range edges is likely to be slowing the rate of spread of Australia's *A. cerana*, relative to a hypothetical population in which all else is equal but no such genetic load occurs.

### Discussion

We tested the prediction that there are two types of founder effects that can shape eco-evolutionary processes in invasive populations: the initial founder event, wherein a limited number of individuals from the parent population arrive in the new location[2] and secondary, serial founder events that occur at range edges as the invasive population expands[10]. The latter can compound the effects of reduced genetic

diversity caused by the former, leading to local increases in homozygosity and genetic load at the advancing range edges. We show that this process is occurring in Australia's expanding invasive *A. cerana* population. At range edges, localized genetic load at the sex locus curbs the reproductive output of colonies and, in turn, the rate at which the population expands. This genetic load arises because of repeated generations of small frontier populations experiencing genetic drift, which overwhelms the negative-frequency-dependent selection acting at *csd*[24] and increases the frequency of the lethal homozygous state.

The localized genetic load documented in our study (arising from increased homozygosity) is just one of several processes that could occur at the edges of expanding populations and impact the rate of spread. In addition, over time, deleterious mutations may accumulate via serial founder effects at the range fronts of expanding populations in a process known as mutation surfing[28,29]. Measurable fitness impacts of such mutation surfing have been previously inferred both for experimental bacterial populations[30,31] and some natural populations whose range expansions were thousands of generations in the past[32,33]. Most notably, the incidence of mildly deleterious mutations among human populations increases with distance from the sub-Sahara[33], consistent with serial founder effects during our species' expansion out of Africa[34,35]. Mutation load at range edges may also interact with the accumulation of deleterious homozygosity and produce a combined effect ('expansion load'[28,29]). Provided mutation load (or expansion load) reduces reproductive output at range edges, such effects presumably also curb the rate of a population's expansion[12].

Tugging in the opposite direction to the expansion load, the conditions at range edges may sometimes instead accelerate the expansion rate via the spatial sorting of high mobility phenotypes[36–38]. This occurs when individuals with traits permitting high dispersal rate co-locate at range edges, leading to assortative mating for those traits and thus offspring with even higher dispersal tendencies. This has been well documented in another Australian invader: the cane toad (*Bufo marinus*). In the continued invasion of cane toads throughout northern Australia, spatial sorting of longer-legged individuals with higher levels of endurance contributes to rapid and accelerating range expansion[37,39]. Whether either of these additional processes (mutation load or spatial sorting) are also occurring in Australia's *A. cerana* remains to be investigated. Presumably, more than one range edge phenomena can interact at the same edge, or processes may differ at different edges of the same expanding population, as well as operating in the same population over different timescales or spatial scales. Thus, the impact of range edge phenomena on the expansion rate in any given invasive population is challenging to predict.

Complementary sex determination is often considered problematic for hymenopteran insects following population bottlenecks as it can increase the risk of extinction for small populations[40]. In general, purging selection helps to counteract founder effects, especially as recessive deleterious alleles are directly exposed to selection in haploid males[41] (provided males express these traits). This is not the case however for sex loci, which are always deleterious in the homozygous state and therefore represent a unique fitness barrier. Even so, some ant, bee and wasp species establish successful invasive populations from limited numbers of founders[25,42–44]. In the case of *A. cerana*, a range of behavioural traits have contributed to mitigating the effect of the initial founder event on nascent invasive populations, including polyandry[45], worker reproduction[23] and cannibalism of diploid male brood[46] (all coupled with strong negative-frequency-dependent selection at *csd*[25]). Other hymenopteran invaders similarly benefit from mitigating reproductive habits that likely aided their establishment[47]. For example, the invasive Red Imported Fire Ant (*Solenopsis invicta*) in the southern U.S.A. has fewer sex alleles and a higher occurrence of diploid males than conspecific populations in native regions[23]. While diploid male incidence in *S. invicta* leads to significantly higher rates of colony

mortality if the colony is monogyne (headed by a single queen), many colonies in the invasive range are instead polygyne (headed by multiple queens, sometimes thousands)[48] and thus incur a lesser cost of diploid male production[49]. Our results highlight however that complementary sex determination can have consequences for the invasion dynamics of hymenopteran populations beyond its initial effect on the probability of establishment, by affecting the rate of spread. Documenting spatial variation in the incidence of diploid males, or its impact on population growth and dispersal, is not practical for many hymenopteran species[50]. We might predict that invader populations supplemented by additional genetic material, either through secondary introductions[51] or subspecies hybridization[52], will experience low rates of diploid males and therefore weak or no impacts on expansion rate. On the other hand, diploid male production at range edges should have particularly strong effects on expansion rates for species which lack the various mitigating reproductive behaviours seen in *Apis* or *S. invicta*; i.e. species in which females mate only once, infertile diploid males survive to reproductive maturity but are sterile or produce sterile offspring, and colonies are monogyne (if the species is social). Indeed, even native-range populations of some Hymenoptera might show reduced fitness at range edges during expansion as a result of complementary sex determination, particularly those with low effective population sizes undergoing range shifts due to climate change, though this possibility remains untested.

In the context of invasive pest management, one of the earliest measures of success for a biological incursion is that the population manages to establish and spread[6]. Here our study highlights that even among such successes, not all invaders are equal. Some loss of genetic diversity relative to native-range populations is common in invaders and will often present no detectable obstacle to growth or spread[21,53,54]. In rare cases, it may even increase levels of additive genetic variance (via the disruption of co-adapted gene complexes) and facilitate adaptation and speciation[55]. However, in the cases where a population's growth or spread is curtailed by inbreeding, its ecological and economic footprint may well also be reduced, relative to populations that grow and spread quickly. This is because slower spread should increase the window available for native biota to respond and for effective management techniques to be developed. In cases of slower-spreading populations, protecting against future incursions of the same species will be particularly important. Australia's *A. cerana* illustrate this well because any secondary incursions in this population are certain to bring new sex alleles[56]. Such supplementation would presumably alleviate the fitness costs of serial founder effects at range edges and thereby accelerate the species' spread across tropical north-eastern Queensland. Until then, while *A. cerana* continues to expand its invasive range in Australia it must do so carrying the baggage of its initial founder event in each outward wave.

## Methods
Specimen collections were made under permit no. WITK18775018 (Queensland Government, Parks and Forests).

### Identifying range edges across time
We used records of *A. cerana* nest collections maintained by Queensland Department of Agriculture and Fisheries (DAF; 2007-2022) to confirm the population was expanding in range over our sampling years and to estimate the location of range edges each year. For these years, we considered an extension of the invasive range to be the first report of a colony in a new locality (suburb or township). We excluded reports within 15 km of Cairns port (the known point of entry, 16°55′57.2″S 145°46′45.0″E) as reports surrounding Cairns port were numerous in the first few years of the incursion and would artificially depress the expansion rate due to false 'first sightings' long after the species had become established in the greater port area. Past June 2013 these reports were less numerous, but still accurately reflect extensions to the invasive range of *A. cerana*. Overall, reports are more

common in the southern part of the range, likely because this region is the most populated and readily accessible to humans. Based on colony detections over the 14-year period since first discovery, we confirm that the population has steadily expanded its range each year, with advancing fronts to the west, north and south. We calculate a mean estimated rate of range expansion of $7.18 \pm 0.78$ km per direction per year ($y = 7.18 \times + 15.54$; $R^2 = 0.73$, $F_{1,32} = 84.9$, $p < 0.001$, distance (km, y) vs time (years, x), linear regression). All statistical tests in this study were performed using R core team 4.0.4.

### Spatial variation in sex locus and neutral loci diversity

Honey bee drones congregate at specific areas (DCAs) at specific times of the day (in Australian *A. cerana*, between 1:30 pm and 3:30 pm AEDT[23]) for the purpose of mating with virgin queens that are attracted to the aggregations. We located DCAs in parks, roadside areas or nature reserves in Far North Queensland within the range of *A. cerana* during the years 2016 (July–September), 2018 (May–June) and 2019 (July–August). To find DCAs we identified candidate locations using Google Maps. We looked for small clearings (<20 m radius), surrounded by trees on three or four sides that were likely to be sheltered from the wind. At suitable DCAs we launched a modified William's drone trap[57] during the *A. cerana* drone flight time (1:00–3:30 pm AEST[23]). This trap relies on drones being attracted to 3–5 queen lures (black cigarette filters) each dosed with 20 μL of Queen Mandibular Pheromone (9-oxo-2-decenoic acid). We aimed to collect at least 100 drones per DCA, which was typically achieved in a single afternoon. Where possible, we visited sites with fewer than 100 drones caught on the first sample day over multiple days to increase sample size. We preserved drones in the field in 100% ethanol.

We aimed to sample DCAs across the population range (Centre, West, and South; Fig. 1), including the northern and southern range edges (Supplementary Table 5). We did not sample drones at the western range edge due to limited road infrastructure in this region. Landscapes to the west also represent a possible climate barrier of increased aridity, while those to the north and south appear ideal habitat for continued expansion of *A. cerana*. Our range edge sampling effort for drones focused particularly on locating DCAs as close as possible to the southern range edge, as the northern edge is in dense rainforest that is not easily accessible. Both northern and southern range edge regions were defined as the area within ~8 km of the putative north and south range edges (indicating the colonies here had likely travelled on the most outward expanding wave of the past 12 months; see "Methods" section above). Non-edge regions (Centre, West and South) included DCAs found further than 8 km from a range edge.

Range edges often have lower population densities than range centres, and so we anticipated that DCAs would be more difficult to locate at range edges. If we sampled a candidate DCA during good weather conditions (>25 °C, <40 km/h wind speed and <3 mm precipitation) and yet failed to detect any drones, we scored this site as 'Drones Absent' (Fig. 1A). Consistent with expectations for true range edges, we found that our nominated range edge regions (Southern and Northern Edges) contained a higher proportion of 'Drone Absent' DCAs than those in the central regions.

We extracted DNA from one hind leg of each drone using the Chelex protocol[58]. We genotyped individuals at a polymorphic fragment of the hypervariable region of *csd* using the protocol developed by Gloag et al.[25]. In this approach, *csd* is amplified using a population-specific set of three primer pairs (Supplementary Table 6). In combination these primers allow the identification of the seven *csd* alleles present in Australia's *A. cerana* population based on differences in allele length (Supplementary Data 1). Allele specificity of *csd* in honey bees, including *A. cerana*, is determined by amino acid differences in the gene's hypervariable region (HVR), with almost all unique specificities varying in nucleotide length[56,59] (e.g., HVR lengths typically vary from 1–21 amino acids, with 97% of alleles varying in length in *A.*

*cerana*[56]). Although functional sex allele mutants should be at a strong selective advantage in this population, we have not detected any alleles with novel lengths in Australia's population, despite extensive genotyping in this and previous studies (>15,183 alleles; any allele lengths different to those of the seven known alleles would reveal themselves as different fragment lengths amplified by one or more of our three primer sets; $N = 2734$[25]; $N = 1920$[45]; $N = 1336$[24]; $N = 9193$, this study).

We performed Monte Carlo simulations to investigate whether regions deviated from population-wide levels of *csd* diversity. First, we calculated the observed haplotype diversity ($H$; i.e. the probability two randomly sampled alleles are different, analogous to heterozygosity in diploids) for *csd* for each region (Fig. 1B). Then, for each region we randomly sampled N sex alleles from the total population dataset, where N was equal to the number of drones actually sampled in that region ($N_{Centre} = 3015$, $N_{South} = 717$, $N_{West} = 627$, $N_{Northern\ Edge} = 129$ and $N_{Southern\ Edge} = 32$). Sex alleles thus had a probability of being sampled in the simulation equal to their frequency in the total population. From this distribution of simulated sex alleles, we then calculated haplotype diversity, and this process was iterated 106 times per region. We considered the haplotype diversity of a region to be significantly different from the simulated distribution if it fell within the highest or lowest 2.5%, highly significantly different if within the highest or lowest 0.5% and very highly significant if within the highest or lowest 0.05% of values (similar to a two-tailed test). As these tests are comparisons of allele frequencies between regions and the total population, they are analogues to a calculation of $F_{st}$. $F_{st}$ itself is not suitable to apply to a locus such as *csd* at which all diploid individuals are heterozygous.

To assess whether the spatial variation in *csd* diversity was also detectable in genome-wide diversity, we used SNP genotyping of drones via the DArTseq reduced-genome sequencing approach, performed by Diversity Arrays Technology Pty Ltd (DArT, Canberra, Australia). This approach achieves complexity reduction prior to sequencing via combinations of restriction enzymes (here *PstI* and *MseI*) that target low-copy genomic regions likely to harbour informative SNPs[60]. The samples we chose for DArTseq spanned the regions of *A. cerana* sampled during 2018 (Centre, Northern Edge, Southern Range Edge, $N = 16$ per region, and South, $N = 15$). We extracted DNA from whole drones using the phenol/chloroform extraction method[61]. Reduced-representation libraries from this DNA were then generated at DArT following the digestion/ligation process described in Kilian et al.[60], except that two restriction enzyme adaptors were used instead of a single *PstI* adaptor. The *PstI* compatible adaptor was designed to include a flow cell attachment sequence (Illumina, San Diego, CA, USA), sequencing primer sequence and barcodes for sample identification (following Elshire et al.[62]). Only fragments that contained both adaptors (*PstI-MseI*) were amplified via PCR (see PCR conditions in Kilian et al.[60]). After PCR, equimolar amounts of each sample were bulked, prior to sequencing on Illumina HiSeq2500 (single-read, 77 cycles, 1.25 M reads per sample). Reads were then processed for SNP identification using DArT's proprietary analytical pipelines to remove low-quality sequences and those with poor repeatability (DArT replicated $N = 24$ samples from DNA digestion through to allelic calls, using independent adaptors, which then serve as technical replicates to confirm repeatability). This preprocessing produced an initial dataset of 6098 SNP markers within 5585 DArT tags (sequences of 69 base pairs each) that mapped to 412 known scaffolds of the *A. cerana* reference genome (ACSNU-2.0[63]) with an average of 14.8 tags per scaffold (Supplementary Data 2–3). We then performed further filtering using the *dartR* package[64] to retain only the highest-quality SNPs. We replaced any heterozygote calls with empty data (as all our samples were haploid) and removed secondary fragments (SNPs that shared a tag). We also retained only those SNPs with a locus call rate threshold >95% (i.e. no more than 5% of samples with missing data) and a reproducibility rate >95% and filtered out minor allele frequencies of less than 2% and SNPs with a Hamming distance <5% (i.e. tag sequences that were more than 95% similar). The final

dataset post-filtering contained 2829 SNPs (mean call rate: 98.5% and reproducibility rate: 99.5%). From this SNP dataset we (i) calculated nucleotide diversity ($\pi$) per region, (ii) visualised differences between regions by performing a PCA (*dudi.pca*; *ade4* package version 1.7-22) and plotting individuals (*ggplot2*) using principal components 1 and 2 (Supplementary Figure 1), and (iii) assessed pairwise genetic differentiation between regions using Nei's $F_{st}$[65] (*pairwise.neifst* & *boot.ppfst*; *hierfstat* version 0.5-11). Finally, we looked for further evidence of lower range edge diversity in neutral loci by genotyping drones ($N_{Centre} = 3110$, $N_{South} = 722$, $N_{Northern\ Edge} = 130$, $N_{West} = 645$ and $N_{Southern\ Edge} = 32$) at eight unlinked microsatellite (single sequence repeat, SSR) loci (A107, Ac1, Ac3, Ac26, Ac27, Ac32, Ac35 and B124[66,67]; primer pairs and PCR conditions in Supplementary Table 6; genotypes in Supplementary Data 1 and Supplementary Table 7). PCR products were electrophoresed on a 3130xl Genetic Analyser and allele calling was performed using GeneMapper 4.0. For microsatellite data, we calculated average haplotype diversity ($H$) per region. To account for differences in sample sizes between regions, we also randomly subsampled each of the central regions (Centre, West and South) to the size of the two range edge regions (Northern Edge and Southern Edge; 131 and 32 samples respectively) from which we recalculated $H$ for each locus. We repeated this process $10^5$ times and obtained 95% confidence intervals such that we could compare the $H$ of central regions to that of range edge regions for each locus. As we did for SNP data, we also performed a PCA (*dudi.pca*; *ade4* package version 1.7-22; plotted using *ggplot2*; Supplementary Figure 1) and calculated the pairwise Nei's $F_{st}$ between regions (*pairwise.neifst* & *boot.ppfst*; *hierfstat* version 0.5-11). To check $F_{st}$ values in this case were not biased by uneven sample sizes, we also estimated 95% confidence intervals of $F_{st}$ values by randomly subsampling 32 drones per region (the sample size of the Southern Edge), repeating this process $10^5$ times. To characterize the power of SSR loci to detect spatial variation in diversity in this population, we also calculated their polymorphic information content (PIC[68]), based on allele richness and frequencies per locus (R core team 4.0.4).

### Brood viability (diploid male production) at range edges

We located 74 colonies via reports from the public to Queensland Biosecurity. These colonies spanned a range of distances from the population range edges of the year of their collection (Supplementary Table 8). Colonies at true range edges are extremely challenging to locate and sample; this is intrinsically so as the density at these edges is very low. This is reflected in our sampling efforts of drones, where range edges had proportionally fewer DCAs that contained drones, and where drones were present, they were fewer in number compared to DCAs in central regions (Fig. 1A). Over this period, six colonies deemed to be at or near range edges were sampled (northern edge: $N = 3$, southern edge: $N = 2$, western edge: $N = 1$) as they were within 8 km of the known range edge, or past this edge, at time of collection, strongly suggesting that they had travelled in the outward expanding wave of the last year.

We extracted the DNA from one hind leg of each worker using the Chelex protocol, as described above for drones and genotyped them at *csd* (range 17–191, mean = 50.3 workers per colony). If the queen had been collected we also extracted her DNA and identified her *csd* genotype directly. Otherwise, we inferred her genotype from the genotypes of her worker offspring (Supplementary Table 8, Supplementary Data 4). If multiple inferred queen genotypes were possible (the case for $N = 14$ colonies), we first determined if there were two dominant alleles in the colony to assign as the queen genotype (ie, two alleles of approximately equal frequency, each present in ~50% of workers, $N = 10$). If this was not the case, we conservatively used the queen genotype that minimised diploid male brood for the colony ($N = 4$). In combination, the genotypes of a queen and her workers can be used to calculate the proportion of diploid male brood in a colony. This is because when a queen produces offspring with a mate that shares one of her two *csd* alleles, 50% of those offspring will be diploid males and the other 50% will be workers that share the queen's genotype at *csd*. As a result, each worker that shares the same genotype as the queen represents an embryo of a diploid male that did not reach adulthood. For each colony, we then calculated the proportion of diploid male brood as;

$$\text{Proportion of diploid male brood} = \frac{d}{n+d}$$

where $n$ is the total sample size of the genotyped workers of the colony and $d$ is the number of workers that shared the queen's genotype (Supplementary Fig. 2).

To establish how diploid male production varied over the invasive range, we calculated the distance between the location of each sampled colony and the population centre (Cairns port, 16°55'57.2" S, 145°46'45.0" E). As population range changes over time and colonies were sampled from multiple years, we standardised distances by dividing them by the estimated range edge at the time of collection. This standardisation allows us to compare how distance from-centre influences demographics without the confounding factor of time. Lastly, we used a linear regression to assess the relationship between proportion of diploid male brood and distance from the range centre.

### Simulations of founder effects at range edges and their impact on expansion rate

We generated agent-based simulations of an invading and dispersing *A. cerana* population with discrete generations[69]. Each generation in the simulations consisted of three stages: (1) a reproductive stage, in which colonies reproduced, (2) a dispersal stage, in which colonies dispersed stochastically either with ($\beta > 0$) or without ($\beta = 0$) a tendency to move to lower density areas of the single continuous spatial dimension, and (3) a persistence stage in which some colonies survived and some died. Each colony was assigned a fitness score according to the proportion of diploid males (DMP) it produced (that is, according to the *csd* genotype of the queen and her mates, where the simulated locus was homozygous-lethal). This fitness score was used to determine, per colony, both the reproductive output and the likelihood of survival until the next generation. Simulated populations started with a low number ($N = 7$) and unequal frequencies of sex alleles, similar to frequencies observed in the real Australian population early during invasion[25]. Where possible, other parameter values were also fixed to match those of Australia's *A. cerana*. Because the precise relationship between diploid male production and fitness is unknown for honey bees, however, we simulated two possible fitness functions: 'Linear' (fitness was negatively proportional to diploid male brood production) and 'Sigmoid' (whereby fitness decreased marginally at low incidences of diploid male production but steeply around incidences of 25%); Supplementary Fig. 3. We also considered a range of values for three other simulation parameters (dispersal strength $\beta = 0$, 5 or 10, average reproductive rate $S_A = 2$, 3 or 4, and maximum population density per unit space $K = 30$, 100 or 300) to confirm that our conclusions were not sensitive to variation in these parameters (see Supplementary Methods 1.1, 2.1). To assess the effect of genetic load at the sex locus on the rate of population spread, we also ran simulations where diploid male production had no fitness cost (the simulated locus was fitness neutral). For every parameter combination, we ran replicate simulations ($N = 96$) for 20 generations (corresponding to ~10–20 years in the real population). Finally, we calculated the maximum distance from the point of incursion reached by populations by generation 20 for each simulation, and the 95% confidence interval of the difference in means of these distances between simulations with or without genetic load at the simulated locus (Supplementary Data 5). A complete description of the model is in Supplementary Methods.

**Reporting summary**

Further information on research design is available in the Nature Portfolio Reporting Summary linked to this article.

## Data availability

The DArTseq raw data files generated in this study have been deposited in the NCBI Sequence Read Archive, BioProject PRJNA1090620. The reference genome used in this study (ACSNU-2.0) is available in GenBank under accession code GCA_001442555.1. The simulation data generated in this study is available at Figshare.com (10.6084/m9.figshare.25395418). All other data supporting the findings of this study are available within the paper and its Supplementary Information and Data (see also Code Availability Statement). Underlying data for Fig. 1A are provided in Supplementary Table 5. Underlying data for Fig. 1B are provided in Supplementary Data 1. Underlying data for Fig. 2A–D are provided in Supplementary Table 8. Underlying Data for Supplementary Fig. 1A are provided in Supplementary Data 2 and 3, and data used to produce Supplementary Fig. 1B are provided in Supplementary Data 1. Summary data for Fig. 3C are included in Supplementary Data 5.

## Code availability

The code used for simulations in this study can be downloaded at https://github.com/Thomas-Hagan/BeeSimulation.git (https://doi.org/10.5281/zenodo.10783818)[69].

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

## Acknowledgements

We would like to acknowledge the traditional custodians of the land on which this fieldwork was performed, the Djabugay, Djiru, Yidinji, Mbarbaram and Kuku Yalanji people, and pay respects to elders past, present and emerging. We thank the Queensland Department of Agriculture and Fisheries, who kindly aided with colony collections in 2018–2021 and provided colony collections for 2012. We also thank Ruby Stephens, Angel Van Bekhoven, Patsavee Utaipanon, Francisco Garcia Bulle Bueno, Jackie McLeod, Beni Cawood and Tony Hagan for assistance during fieldwork. This work was funded by an Australian Research Council grant DP190101500 (R.G., B.P.O.).

## Author contributions

T.H., G.D., R.G. and B.P.O. collected field data. T.H., G.D. and G.B. conducted lab work. T.H. analysed the data and wrote the simulation code. T.H., R.G. and B.P.O. designed the study. T.H. and R.G. wrote the manuscript. All authors provided feedback on the data and manuscript.

## Competing interests

The authors declare no competing interests.
