## [Peer Review File · Nature Communications]

Serial founder effects slow range expansion in an invasive social insectREVIEWER COMMENTS

Reviewer #1 (Remarks to the Author):

This is an exciting empirical population genomic study that shows very nicely how loss of genetic diversity in an invasive population that has undergone a population bottleneck during initial founding can be further exacerbated at the geographical margins of the invasive range. Importantly, the authors show that there are demonstrable negative fitness consequences of this secondary loss of diversity, due to the special mode of sex determination of honey bees. The study was well planned and executed, with appropriate sampling of the bees and use of different types of molecular genetic markers, it features great sample sizes and appropriate methods of analysis of the data, and it is very well written. I have only a few, relatively minor, comments for the authors to consider as they revise the manuscript.

line 31: The authors might clarify that reductions in genetic diversity and changes in allele frequencies following a founder event have several causes, including loss of alleles when the founder group size is insufficient to carry the total number of alleles at a locus (highly likely with CSD in invasive Hymenoptera), loss of alleles by stochastic sampling effects even when the founder number is large enough to carry all the alleles at a locus, and drift for several generations after the founder event due to continued low effective population size.

lines 193-194: Only traits expressed in males are subject to such selection, so presumably much of the genome, including many genetic elements underlying sex-limited traits, are shielded from this process (although not CSD).

Line 254: This seems to be strange placement for this sentence; does it refer just to the statistics described in this paragraph?

Line 272: Why use the trinomial here suddenly? It is used nowhere else.

Line 334: "these data"

Fig. 1A: There is plenty of white space on the map to the west of the invasive range to display an inset showing the study area relative to the entire continent; many folks will not know where the Cairnes coast is relative to other major geographic features.

Discussion: I was surprised that more connections were not made between the example of the study population and the example of the invasive populations of the fire ant *Solenopsis invicta* in the U.S., because there are many potential points of comparison between these two highly eusocial hymenopterans with CSD. Importantly, the fire ant studies include a clear demonstration of the fitness consequences of loss of CSD alleles (and the causes of the diminished fitness) under controlled laboratory conditions. Although the genetic toolbox was quite meager in most of the early studies by today's standards, sample sizes were large for surveys of several widely separated populations in the U.S. Paradoxically, a population close to the site of invasion of the U.S. by *S. invicta* (Mississippi Population) seemed to have fewer CSD alleles than a known peripheral population that had been out of contact with the main population for some time (Georgia population).

Also, I was surprised that there was little mention of the considerable body of theoretical and empirical research on ancient and recent serial founder effects, especially in humans, that might be drawn upon to broaden the Discussion. Authors of this material, much of which focusses on the role of such populations in speciation, include the likes of Mayr, Carson, Futuyma, and the Grants, and these studies concern species across the tree of life.

Reviewer #2 (Remarks to the Author):

This paper explores serial founder effects at the edge of an expanding population and their influence on viability in *Apis cerana* in Australia.

The authors rightly describe the invasion and expansion of *A. cerana* in Australia as an excellent model to investigate the effects of reduced genetic diversity and increased drift at population edges, in particular of expanding populations. While this paper unfortunately falls short on direct measures of nucleotide diversity and deleterious mutations, it takes advantage of the haplodiploid sex determining system in these bees, which is based on *csd*. Using *csd* alleles, one can determine the frequency of fitness-relevant trait of diploid male production and hence can link genetic diversity (from SSRs/SNPs/*csd* alleles) with fitness (diploid male production). This is an elegant approach and can be used nicely to show how fitness is reduced at population edges. It is unfortunate that no whole genome data can show frequency of other deleterious mutations which may be fitness relevant too. Despite this shortcoming, the paper provides a good case for the serial founder effects in an expanding population. One problem the study has are the very low sample sizes for some cases on the population edge and also the very large sample size differences between some areas/groups. While *csd* allele diversity and DMP production are good measures, I am not so convinced that 8 SSRs and >1500 SNPs provide a sufficient basis for the claims. Only PCA was used to explore the differentiation (or lack thereof) between central range and range edge populations (but not corrected for sample size differences). Perhaps calculating nucleotide diversity and allele numbers/frequencies (or allelic richness for SSRs) and hence an indirect insight into heterozygosity across the population would be interesting to explore. Such data could then hopefully support the results.

I have a few more comments below. A more general comment is that the paper somewhat seems to lack a broader integration into the literature, e.g. discussing general aspects / citing relevant papers regarding range expansions / population edges and their population genomic impact. I am also missing a comparison/discussion with another honeybee invasion, the spread of the Africanized honeybee in Latin America towards the north and the south. That had been investigated using whole genome data and may provide important additional insights (Calfee ... Coop, PlosGen 2020) and is not even cited here. In general I had the impression that not a lot of papers are referenced for many of the statements made. I would suggest to improve this.

L44 Too generalized. Some Hym have no *csd* (eg *Nasonia* = imprinting). Citations are missing.

L93 Confusing sentence. Says reduction was modest/severe in North/South but then says 0.25/0.02 lower than center.

L94: same sizes are very different. Have these PCA comparisons been corrected by sample size, or tried to replicate the results with a dataset where one group is downsampled to be identical to the other group.

L106 Here u refer to methods (should be "Materials and Methods") but should also refer to suppl fig S5. Also regarding this: supplementary methods list the figure as "Supplementary Fig. S5" while in the main text it is referred to as "Fig S4, supplementary methods"

L118 DMP - the abbreviation needs to be introduced in the first part of the sentence

L117/119 the difference in DMP is 16 vs 20%. Is this a big difference?

L194 exposure in haploids still may not occur if the gene /exon/splice variant of interest is not expressed in males.

L272 here the subspecies *A.c.javana* is mentioned for the first time. Either this should also be mentioned early in the MS or be left out.

L288. The mentioned citation is not reference properly (number).

L320: information on the reduced representation sequencing is missing (which enzyme and protocol, sequenced at which depth, read length platform, filter steps, which tools were used, based on which mapping/reference etc)

L340 add more reasoning here and don't just simple statements only

-add more specifics of your methods: eg L264: what pheromone and quantity, L288: primers/annealing temp etc of the PCR test. Don't have the reader look this up elsewhere where it is perhaps inaccessible. Specify the necessary things here. L318-326 This whole section is missing a lot of relevant information: e.g. what reference was used. What read filters. What was used for mapping and then SNP calling and what were parameters filters and cutoffs and which software

(versions) were used.

Fig2D

There are some more outliers in closer non edge colonies. Can you provide more information/interpretation on this?

There is no data availability paragraph. The reporting form alone is insufficient. The main sources should be specified in the main MS. Is the SNP genotype, SSR genotype and csd genotype data available somewhere? Edit: nevermind, i found it in an suppl file. It contains also info on the primers (see my above comment). It would be worth referencing this suppl table in the relevant places.

Referencing of the supplement materials needs to be made consistent, sometimes called suppl Methods and sometime suppl material or materials, sometimes the referenced Figure is stated first sometimes last. The first suppl fig mentioned is S3A, then suppl. Materials 1.1 and 2.1, then fig S4, S1, i.e. the order is weird.

RESPONSE TO REVIEWERS' COMMENTS

Reviewer #1 (Remarks to the Author):

This is an exciting empirical population genomic study that shows very nicely how loss of genetic diversity in an invasive population that has undergone a population bottleneck during initial founding can be further exacerbated at the geographical margins of the invasive range. Importantly, the authors show that there are demonstrable negative fitness consequences of this secondary loss of diversity, due to the special mode of sex determination of honey bees. The study was well planned and executed, with appropriate sampling of the bees and use of different types of molecular genetic markers, it features great sample sizes and appropriate methods of analysis of the data, and it is very well written. I have only a few, relatively minor, comments for the authors to consider as they revise the manuscript.

line 31: The authors might clarify that reductions in genetic diversity and changes in allele frequencies following a founder event have several causes, including loss of alleles when the founder group size is insufficient to carry the total number of alleles at a locus (highly likely with CSD in invasive Hymenoptera), loss of alleles by stochastic sampling effects even when the founder number is large enough to carry all the alleles at a locus, and drift for several generations after the founder event due to continued low effective population size.

Reply: Thank you. This is now clarified in the Introduction as follows:

“A small founding population may not reflect the allelic composition of its source population because founding individuals fail to transfer all source alleles to the new population and/or because founder allele frequencies differ strongly from those of the source population³. These effects are then exacerbated by genetic drift in the generations immediately following the founding event⁴.” (lines 32-35)

lines 193-194: Only traits expressed in males are subject to such selection, so presumably much of the genome, including many genetic elements underlying sex-limited traits, are shielded from this process (although not CSD).

Reply: That's correct, thank you. We have revised for clarity as:

“In general, purging selection helps to counteract founder effects, especially as recessive deleterious alleles are directly exposed to selection in haploid males⁴⁰ (provided males express these traits).” (lines 220-221)

Line 254: This seems to be strange placement for this sentence; does it refer just to the statistics described in this paragraph?

Reply: We agree this was awkwardly placed. We have revised here to state *“All statistical tests in this study were performed using R core team 4.0.4.”* (line 287) and we refer to specific packages throughout the Methods as relevant.

Line 272: Why use the trinomial here suddenly? It is used nowhere else.

Reply: Thank you for picking this up. We have revised to the binomial *A. cerana* (line 307).

Line 334: "these data"

Reply: Thank you, this entire paragraph has been reworked (lines 354-406).

Fig. 1A: There is plenty of white space on the map to the west of the invasive range to display an inset showing the study area relative to the entire continent; many folks will not know where the Cairnes coast is relative to other major geographic features.

Reply: That's an excellent suggestion thank you. We have added an inset map of Australia to Figure 1A.

Discussion: I was surprised that more connections were not made between the example of the study population and the example of the invasive populations of the fire ant *Solenopsis invicta* in the U.S., because there are many potential points of comparison between these two highly eusocial hymenopterans with CSD. Importantly, the fire ant studies include a clear demonstration of the fitness consequences of loss of CSD alleles (and the causes of the diminished fitness) under controlled laboratory conditions. Although the genetic toolbox was quite meager in most of the early studies by today's standards, sample sizes were large for surveys of several widely separated populations in the U.S. Paradoxically, a population close to the site of invasion of the U.S. by *S. invicta* (Mississippi Population) seemed to have fewer CSD alleles than a known peripheral population that had been out of contact with the main population for some time (Georgia population).

Reply: Thank you for this comment. We agree the case of *S. invicta* is a particularly interesting and well-documented case with points of comparison to *A. cerana*. We now highlight the invasion of *S. invicta* in the Discussion as an example in which genetic load at the sex locus was significant after the initial founder event (lines 227-234). It is also an interesting comparison in that certain mitigating behaviours appear to have aided its establishment (polygyny), as they have likely done also in *A. cerana* (polyandry).

And the following citations on *S. invicta* added:

Ross KG, Fletcher DJC. (1985). Genetic origin of male diploidy in the fire ant, *solenopsis invicta* (hymenoptera: formicidae), and its evolutionary significance. *Evolution* **39**, 888-903

Ross K, Fletcher D. (1986). Diploid male production - a significant colony mortality factor in the fire ant *Solenopsis invicta* (Hymenoptera: Formicidae). *Behavioral Ecology and Sociobiology* **19**, 283-291

Ross KG, Shoemaker DD. (2008). Estimation of the number of founders of an invasive pest insect population: the fire ant *Solenopsis invicta* in the USA. *Proceedings of the Royal Society B* **275**, 2231-2240

In addition to the below study, previously cited:

Ross KG, Vargo EL, Keller L, Trager JC. (1993). Effect of a founder event on variation in the genetic sex-determining system of the fire ant *Solenopsis invicta*. *Genetics* **135**, 843-854

Also, I was surprised that there was little mention of the considerable body of theoretical and empirical research on ancient and recent serial founder effects, especially in humans, that might be drawn upon to broaden the Discussion. Authors of this material, much of which focusses on the role of such populations in speciation, include the likes of Mayr, Carson, Futuyma, and the Grants, and these studies concern species across the tree of life.

Reply: Thank you for this comment. In response we have expanded our discussion of previous empirical findings that have demonstrated genetic load associated with serial founder effects at the expanding edges of populations, including in humans (line 194-199).

With respect to the literature on founder effects and speciation (often focused on island systems), we are familiar with the work and agree it is insightful. We now mention this work briefly in the Discussion to acknowledge that founder effects can promote genetic innovation under some conditions (lines 255-257; Carson 1990) and cite Mayr's original formulation of the founder effect in the Introduction (line 31). However, our view is that further expanding our discussion to cover the speciation literature in depth would detract from the focus of the question and conclusions of our present study, which is an understanding of the short-term impacts of range edge founder effects on invasive population dynamics.

Carson HL. Increased genetic variance after a population bottleneck. *Trends in Ecology & Evolution* **5**, 228-230 (1990).

Mayr E. *Systematics and the Origin of Species.* Columbia University Press (1942).

Reviewer #2 (Remarks to the Author):

This paper explores serial founder effects at the edge of an expanding population and their influence on viability in *Apis cerana* in Australia.

The authors rightly describe the invasion and expansion of *A. cerana* in Australia as an excellent model to investigate the effects of reduced genetic diversity and increased drift at population edges, in particular of expanding populations. While this paper unfortunately falls short on direct measures of nucleotide diversity and deleterious mutations, it takes advantage of the haplodiploid sex determining system in these bees, which is based on *csd*. Using *csd* alleles, one can determine the frequency of fitness-relevant trait of diploid male production and hence can link genetic diversity (from SSRs/SNPs/*csd* alleles) with fitness (diploid male production). This is an elegant approach and can be used nicely to show how fitness is reduced at population edges. It is unfortunate that no whole genome data can show frequency of other deleterious mutations which may be fitness relevant too. Despite this shortcoming, the paper provides a good case for the serial founder effects in an expanding population. One problem the study has are the very low sample sizes for some cases on the population edge and also the very large sample size differences between some areas/groups. While *csd* allele diversity and DMP production are good measures, I am not so convinced that 8 SSRs and >1500 SNPs provide a sufficient basis for the claims. Only PCA was used to explore the differentiation (or lack thereof) between central range and range edge populations (but not corrected for sample size differences). Perhaps calculating nucleotide diversity and allele numbers/frequencies (or allelic richness for SSRs) and hence an indirect insight into heterozygosity across the population would be interesting to explore. Such data could then hopefully support the results.

Reply: Thank you for the helpful comments. As suggested, we have now included additional information and analyses relating to our dataset of SNPs and SSRs (proxies of neutral genome-wide genetic diversity). In particular, we now present:

1. Nucleotide diversity (π) and a measure of genetic differentiation between regions (F_{st}) for our SNP dataset. Overall, π is low in this population (as expected given the initial founder effect) and further declines towards the range edges. There is also increasing differentiation between subpopulations with distance (based on pairwise F_{st}) consistent with range edge founder effects. (Results, lines 100-111).
2. Haplotype diversity (H), polymorphic information content (PIC) and differentiation (pairwise F_{st}) for each of eight SSRs (microsatellites). In this population, these loci have low polymorphism in all regions due to the initial founder event and, unfortunately therefore, limited power to detect spatial differentiation in diversity

(average PIC = 0.31). We found some loci consistent with lower H at range edges at the Southern Edge, but no such effects at the Northern Edge (indeed, some SSRs had higher H at that edge). H values here are based on subsampling simulations to account for sample size differences (see revisions to Methods lines 393-406). Pairwise F_{st} (and subsampled pairwise F_{st}) for these loci was consistent with differentiation between regions, but showed very large confidence intervals, often overlapping with 0.

In sum therefore, these new analyses support our original conclusion that neutral diversity is lower at range edges than range centre, in addition to the lower range edge diversity at the fitness-critical *csd* locus shown in Figure 1. As indicated in our initial manuscript, the SSRs (microsatellites) mostly fail to identify population substructure due to low power (a result of the severity of the initial bottleneck removing most of the polymorphism at SSRs), but our SNP dataset is sufficient to detect this spatial differentiation.

Further responses:

Related to our SNP dataset, we also revisited our filtering approach and increased the final number of SNPs retained in the analysis from 1500 to 2869. Note that the overall conclusions do not change between the two sets (that is, using the initial set of 1500 SNPs we also find declining nucleotide diversity with distance to range edge, increasing F_{st} with distance, and PCA with range edge outliers), however we feel that the revised filtering achieves a better balance between retaining information and removing non-independent SNPs. Specifically, we had originally used a very strict Hamming Distance filter (retaining only SNPs from DArT tags if tags had < 80% sequence similarity). This filter is intended to remove SNPs in regions that might be duplicated due to transposable elements or other historical genome duplications. Such events are unlikely to have impacted this population on such a short timescale (< 20 years). We have therefore now set this filter at 95% (tags with <95% sequence similarity retained). All other filters remain at their original conservative values (locus call rate threshold >95%, reproducibility rate >95%, minor allele frequencies > 2%; Methods, lines 376-382).

We agree that whole genome data from different regions of this expanding invasive population would be an interesting dataset for future work and could potentially identify novel mutations that have appeared at range edges. Confirming whether an identified mutation is deleterious in such a dataset, however, is often challenging. For this reason, we chose to focus the current study on the unique opportunity presented by *csd*, a locus for which we can *directly link* lower diversity to lower fitness (thanks to the fantastic work of many previous researchers, whose efforts have characterized the function of this critical gene). Note also that whole genome sequencing (if short read) cannot capture *csd* allele diversity because of the nature of this locus (different alleles have varying numbers of repeats of the same motifs in a hypervariable region) – genotyping at *csd* therefore requires a targeted sequencing approach, or prior characterization of all alleles in the population followed by allele-specific genotyping (i.e. the approach we used).

Finally, with respect to the low sample sizes on the population edge relative to the population centre for the Part 1 dataset of our study, these sample size differences are challenging to avoid given the intrinsically low density of individuals at true expanding range edges. This was demonstrated in our fieldwork by the number of suitable drone congregation areas (DCAs) at range edges at which we failed to detect drones, despite a high sampling effort, and the low number of drones per DCA at range edges (see Methods and Figure 1A). We nevertheless have confidence in our Part 1 conclusions because we: (1) performed bootstrap Monte Carlo methods of *csd* diversity to account for sample size differences (Figure 1B), (ii) find consistent patterns of lower diversity at range edges for both *csd*, and SNPs, and (iii) collected and analysed the independent datasets from Part 2 (colony-level DMP as a function of distance from the range centre) and Part 3 (simulations of

expanding honey bee populations under a range of conditions), which support the conclusion from Part 1 that *csd* diversity is lower at range edges than the range centre.

I have a few more comments below. A more general comment is that the paper somewhat seems to lack a broader integration into the literature, e.g. discussing general aspects / citing relevant papers regarding range expansions / population edges and their population genomic impact. I am also missing a comparison/discussion with another honeybee invasion, the spread of the Africanized honeybee in Latin America towards the north and the south. That had been investigated using whole genome data and may provide important additional insights (Calfee ... Coop, PlosGen 2020) and is not even cited here. In general I had the impression that not a lot of papers are referenced for many of the statements made. I would suggest to improve this.

Reply: We have revised our Discussion and integrated references to additional literature throughout our manuscript, resulting in an increase from 46 to 67 total citations. Our revised Discussion now includes discussion of invasive fire ants in the US, another invasion by a social hymenopteran that has many points of comparison with diploid male production (lines 227-234), and elaborates on other key studies on the dynamics of expanding populations (lines 194-212). We have also carefully checked all our statements to ensure appropriate citations in all cases (see revised highlighted manuscript).

We have opted not to include an in-depth discussion of the invasion of Africanized honey bees into the Americas however (e.g. Calfee et al 2020), despite this being a very interesting invasion. This invasive sub-species of the western honey bee (i.e. the Africanised honey bee *Apis mellifera scutellata*) was mating freely with the *Apis mellifera mellifera* already present in the Americas throughout its invasion. While it is therefore a fascinating system for studying hybridization and introgression, that *A. mellifera* population never faced low densities at range edges or low sex allele numbers during its spread. However, we have now included Calfee et al 2020 as a reference to how hybridisation may impact range edge founder effects, during our discussion on complementary sex determination (line 240).

Calfee E, Agra MN, Palacio MA, Ramírez SR, Coop G. Selection and hybridization shaped the rapid spread of African honey bee ancestry in the Americas. PLOS Genetics **16**, e1009038 (2020).

L44 Too generalized. Some Hym have no *csd* (eg *Nasonia* = imprinting). Citations are missing.

Reply: We have revised as follows:

“Invasive hymenopterans in the clade Aculeata (the ants, bees, and stinging wasps), however, provide an opportunity to do so due to their system of sex determination. Hymenopterans are haplodiploid: males are haploid, females are diploid. Yet in many species, sex is ultimately determined by zygosity at one or more “sex loci” in a system known as Complementary Sex Determination^{15,16}. All haploid individuals develop as normal males and diploid individuals develop as females if they are heterozygous at the sex locus (or at least one sex locus if there are multiple). Diploid individuals that are homozygous at the sex locus (or loci) instead develop as “diploid males” that are infertile, inviable or cannibalised as larvae¹⁵.” (lines 48-56)

L93 Confusing sentence. Says reduction was modest/severe in North/South but then says 0.25/0.02 lower than center.

Reply: We have revised to avoid confusion, as follows:

“This reduction in csd haplotype diversity (the result of a shift in allele frequencies relative to the central region), was modest at the Northern Edge ($H = 0.82$ vs 0.85 in the range centre) but severe at the Southern Edge ($H = 0.56$).” (lines 96-99)

L94: same sizes are very different. have these PCA comparisons been corrected by sample size, or tried to replicate the results with a dataset where one group is downsampled to be identical to the other group.

Reply: The PCA based on our SNP dataset (Supp Figure 1A) uses a dataset with equal sample sizes per region (N=15-16 drones per region) and indicates differentiation of both the southern and northern edges relative to the central regions. The PCA based on our SSR dataset (Supp Figure 1B) uses a dataset with unequal sample sizes and shows no clear differentiation between regions. However, as detailed above, we now include a range of additional analyses on both datasets beyond just the PCAs (Results, lines 101-109, Methods 386-411, and Supp Tables 1-4, Supp Data 6), including analyses of the SSR data that accounts for the unequal sample sizes (Haplotype diversity estimates and confidence intervals for each region when downsampled to match edge region sample sizes; see Supp Table 3).

L106 Here u refer to methods (should be "Materials and Methods") but should also refer to suppl fig S5. Also regarding this: supplementary methods list the figure as "Supplementary Fig. S5" while in the main text it is referred to as "Fig S4, supplementary methods"

Reply: Thank you. We have carefully checked all formatting in accordance with the journal's requirements, and checked all mentions of Supplementary Information Figures, Tables and Data for consistency.

L118 DMP - the abbreviation needs to be introduced in the first part of the sentence

Reply: We have rephrased so that this sentence does not use the DMP abbreviation.

LI117/119 the difference in DMP is 16 vs 20%. Is this a big difference?

Reply: The average DMP for the range centre and range edge was 16% and 35% respectively (a difference of 18% diploid brood viability, an effective doubling). We also highlight here that all of the six most extreme range edge colonies had high DMP (minimum 20% to maximum 50%), while most centre colonies did not reach such high values. We have rephrased these sentences for better clarity:

“Thus while colonies with high diploid male production still occurred in the population centre, on average, those colonies closer to the range edges had higher proportions of inviable diploid male brood (Figure 2D). Indeed, all six colonies sampled at or very near the most extreme range edges (northern edge: $N = 3$, southern edge: $N = 2$, western edge: $N = 1$) had diploid male production in the range 20 - 50% (i.e. at least one in every five diploid embryos were inviable, and as much as one in every two; mean proportion of diploid male brood for edge and non-edge colonies = 35% and 16% respectively).” (Results, lines 131-138)

L194 exposedm in haploids still may not occur if the gene /exon/splice variant of interest is not expressed in males.

Reply: We have revised this sentence as follows:

“In general, purging selection helps to counteract founder effects, especially as recessive deleterious alleles are directly exposed to selection in haploid males⁴⁰ (provided males express these traits).” (lines 220-221)

L272 here the subspecies *A.c.javana* is mentioned for the first time. Either this should also be mentioned early in the MS or be left out.

Reply: Thank you for picking this up. We have revised to *A. cerana*.

L288. The mentioned citation is not reference properly (number).

Reply: We have added the citation number (line 324).

L320: information on the reduced representation sequencing is missing (which enzyme and protocol, sequenced at which depth, readlength platform, filter steps lack which tools were used, based on which mapping/reference etc)

Reply: We have added more information on the DArTseq reduced genome sequencing in Methods. Note that the primary filtering (initial filtering for high sequence quality, repeatability and SNP calling) is performed by DArT using proprietary pipelines. The DArTseq approach uses *de novo* SNP calling and does not rely on genome mapping, although the SNPs in our case were subsequently mapped to the *A. cerana* reference genome to confirm their distribution across scaffolds. Methods relating to this data now read:

*“To assess whether the spatial variation in *csd* diversity was also detectable in genome-wide diversity, we used SNP genotyping of drones via the DArTseq reduced-genome sequencing approach, performed by Diversity Arrays Technology Pty Ltd (DArT, Canberra, Australia). This approach achieves complexity reduction prior to sequencing via combinations of restriction enzymes (here *Pst*I and *Mse*I) that target low-copy genomic regions likely to harbour informative SNPs⁵⁷. The samples we chose for DArTseq spanned the regions of *A. cerana* sampled during 2018 (Centre, Northern Edge, Southern Range Edge, N = 16 per region, and South, N = 15). We extracted DNA from whole drones using the phenol/chloroform extraction method⁵⁸. Reduced-representation libraries from this DNA were then generated at DArT following the digestion/ligation process described in Kilian et al. 2012⁵⁷, except that two restriction enzyme adaptors were used instead of a single *Pst*I adaptor. The *Pst*I compatible adaptor was designed to include a flow cell attachment sequence (Illumina, San Diego, CA, USA), sequencing primer sequence and barcodes for sample identification (following Elshire et al. 2011⁵⁹). Only fragments that contained both adaptors (*Pst*I-*Mse*I) were amplified via PCR (see PCR conditions in Kilian et al. 2012⁵⁷). After PCR, equimolar amounts of each sample were bulked, prior to sequencing on Illumina HiSeq2500 (single-read, 77 cycles, 1.25M reads per sample). Reads were then processed for SNP identification using DArT’s proprietary analytical pipelines to remove low quality sequences and those with poor repeatability (DArT replicated N=24 samples from DNA digestion through to allelic calls, using independent adaptors, which then serve as technical replicates to confirm repeatability). This preprocessing produced an initial dataset of 6098 SNP markers within 5585 DArT tags (sequences of 69 base pairs each) that mapped to 412 known scaffolds of the *A. cerana* reference genome (ACSNU-2.0⁶⁰) with an average of 14.8 tags per scaffold (Supplementary Data 4-5). We then performed further filtering using the *dartR* package⁶¹ to retain only the highest quality SNPs. We replaced any heterozygote calls with empty data (as all our samples were haploid) and removed secondary fragments (SNPs that shared a tag). We also retained only those SNPs with a locus call rate threshold >95% (i.e. no more than 5% of samples with missing data) and a reproducibility rate >95%, and filtered out minor allele frequencies of less than 2% and SNPs with a Hamming distance <5% (i.e. tag sequences that were more than 95% similar). The final dataset post-filtering contained 2829 SNPs (mean call rate: 98.5% and reproducibility rate: 99.5%).” (lines 354-385)*

L340 add more reasoning here and don't just simple statements only

Reply: We appreciate the call for clarity surrounding this statement. The low density at range edges is supported by both our drone collection observations (in which we failed to find drones at many suitable congregation areas at range edges, despite high sampling efforts) and our subsequent simulation results (see Figure 3B). We therefore revise this section to now read:

“Colonies at true range edges are extremely challenging to locate and sample; this is intrinsically so as the density at these edges is very low. This is reflected in our sampling efforts of drones, where range edges had proportionally fewer DCAs that contained drones, and where drones were present they were fewer in number compared to DCAs in central regions (Figure 1A).” (lines 410-414)

-add more specifics of your methods: eg L264: what pheromone and quantity , L288: primers/annealing temp etc of the PCR test. Dont have the reader look this up elsewhere where it is perhaps inaccessible. Specify the necessary thing here. L318-326 This whole section is missing alot of relevant information: e.g. what reference was used. What readfilters. What was used for mapping and then SNP calling and what were parameters filters and cutoffs and which software (versions) were used.

Reply: We have added the pheromone name and quantity (line 298), primer names and annealing temps are given in Supplementary Data 2, and details on the DArTseq approach and analysis as now given, as described above (lines 354-385).

Fig2D

There are some more outliers in closer non edge colonies. Can you provide more information/interpretation on this?

Reply: Thank you for this comment. Yes, some colonies close to the range centre nevertheless have high diploid male production (DMP) through random chance. That is, being a range-centre colony is not a guarantee of low DMP in this population, as the original founder event left it with only seven sex alleles (a fraction of the diversity found in native range populations). Nevertheless, as Fig 2D shows, we find a strong relationship between DMP and distance from the range centre, such that, on average, range edge colonies are more likely to have higher DMP than those in the range centre. We have added the following text to the Results where Figure 2D is introduced to better clarify this point:

“Thus while colonies with high diploid male production still occurred in the population centre, on average, those colonies closer to the range edges had higher proportions of inviable diploid male brood (Figure 2D).” (Results, lines 131-133)

There is no data availability paragraph. The reporting form alone is insufficient. The main sources should be specified in the main MS. Is the SNP genotype, SSR genotype and *csd* genotype data available somewhere? Edit: nevermind, i found it in an suppl file. It contains also info on the primers (see my above comment). It would be worth referencing this suppl table in the relevant places.

Reply: We apologise for the confusion caused here. Our manuscript now contains a Data Availability Statement and Code Availability Statement. All the raw data from our study is provided in Supplementary Data, including the full set of *csd*, SSR and SNP genotypes. The code to generate all the simulation data can be downloaded from GitHub. We have added further references to Supplementary Data at relevant places throughout the main text (see highlighted mark-up of revised manuscript).

Referencing of the supplement materials needs to be made consistent, sometimes called suppl Methods and sometime suppl material or materials, sometimes the referenced Figure is stated first sometimes last. The first suppl fig mentioned is S3A, then suppl. Materials 1.1 and 2.1, then fig S4, S1, i.e. the order is weird.

Reply: Thank you for picking this up. We have corrected and checked all mentions of Supplementary Information and they are now consistent throughout the manuscript and in line with the formatting guidelines for the journal (see highlighted mark-up of revised manuscript).

REVIEWERS' COMMENTS

Reviewer #1 (Remarks to the Author):

The authors have acknowledged the points made in the previous reviews and modified the current version accordingly. This new version reads very well, I think, and it should now be ready for publication in Nature Communications.

Reviewer #2 (Remarks to the Author):

The revisions provided by the authors addressed the critical issues identified earlier. The manuscript hence is much improved. In particular the additional analyses of nucleotide and haplotype diversity are helpful and support the initial pattern. Apart from low samples size at population edges - which are inevitably hard to sample - all issues were clarified/addressed.